# A possible route towards dissipation-protected qubits using a multidimensional dark space and its symmetries

Raul A. Santos [1]✉, Fernando Iemini[2,3], Alex Kamenev[4,5] & Yuval Gefen[6]

Quantum systems are always subject to interactions with an environment, typically resulting in decoherence and distortion of quantum correlations. It has been recently shown that a controlled interaction with the environment may actually help to create a state, dubbed as "dark", which is immune to decoherence. To encode quantum information in the dark states, they need to span a space with a dimensionality larger than one, so different orthogonal states act as a computational basis. Here, we devise a symmetry-based conceptual framework to engineer such degenerate dark spaces (DDS), protected from decoherence by the environment. We illustrate this construction with a model protocol, inspired by the fractional quantum Hall effect, where the DDS basis is isomorphic to a set of degenerate Laughlin states. The long-time steady state of our driven-dissipative model exhibits thus all the characteristics of degenerate vacua of a unitary topological system.

[1] T.C.M. Group, Cavendish Laboratory, University of Cambridge, J.J. Thomson Avenue, Cambridge CB3 0HE, UK. [2] Instituto de Física, Universidade Federal Fluminense, Niterói 24210-346, Brazil. [3] Abdus Salam ICTP, Strada Costiera 11, Trieste I-34151, Italy. [4] School of Physics and Astronomy, University of Minnesota, Minneapolis, MN 55455, USA. [5] William I. Fine Theoretical Physics Institute, University of Minnesota, Minneapolis, MN 55455, USA. [6] Department of Condensed Matter Physics, The Weizmann Institute of Science, Rehovot 76100, Israel. ✉email: rs2000@alumni.cam.ac.uk

t is believed that dissipation conspires against coherence of quantum states, rendering them to be close to a classical ensemble. This belief was recently challenged by approaches aimed at incorporating both drive and dissipation to reach a correlated coherent steady state[1–11]. One remarkable example has been the idea of harnessing dissipation to purify nontrivial topological states[12–14]. This is achieved by a careful interplay between radiation-induced drive, and coupling to an external bath, that provides a desired relaxation channel. A sequence of excitations and relaxations generates, in the long time limit, a non-equilibrium steady state that decouples from the external drive creating a decoherence free subspace[15,16] dubbed a dark state. This idea opens a way to engineer a rich variety of non-trivial stationary states, going well beyond thermal states of equilibrium systems.

In order to use this approach to design (and ultimately manipulate) qubits, it is necessary to engineer a non-equilibrium steady space, which is at least two dimensions[14,17]. Here we develop a framework to construct driven-dissipative schemes with degenerate dark spaces (DDS). We achieve this goal by analyzing the role of symmetries in dissipative dynamics. Specifically, we claim that the dimensionality of DDS is given by the period of the projective symmetry representation[18], inherent to the system's evolution (which is considered to be Lindbladian[19,20]). To this end we extend the discussion of Lindbladian symmetries[21–23] to include those that are realized projectively, providing a link between the projective representations and the dimensionality of the DDS density operator.

We illustrate this framework by studying driven-dissipative evolution of a correlated one-dimensional (1D) system, inspired by Laughlin quantum Hall states with $v = 1/m$ filling fractions ($m$ is an odd integer) in a quasi-1D strip (the so-called "thin torus limit"). This evolution is described by a Lindbladian master equation that possesses a DDS. The latter is spanned by $m$ orthogonal vectors, isomorphic to the set of many-body Laughlin ground states on the torus. This correlated DDS has an extra advantage of being exactly described by computationally convenient matrix product states (MPS). We design a systematic protocol, based on adiabatically varied Lindbladians, that maximizes the purity and fidelity (overlap with the dark space) of its ultimate steady states.

## Results

### Projective representation of symmetries in Hamiltonian dynamics.
As a warm up for the symmetry discussion, let us consider the Hamiltonian case. If a Hamiltonian is invariant under the action of a symmetry group $\mathcal{G}$, then action of the group elements, $g \in \mathcal{G}$, on a state is implemented by a unitary representation[18]. In particular, the eigenstates of the Hamiltonian can be labeled by eigenvalues of the symmetry operator. As quantum states form rays in the Hilbert space, such that states $|\psi\rangle$ and $e^{i\phi}|\psi\rangle$ are equivalent, it is natural to consider representations that satisfy the group multiplication rule up to a phase, i.e. projective representations[18], defined as

$$D(g_1)D(g_2) = e^{i\phi(g_1,g_2)}D(g_1g_2), \quad (1)$$

where $D(g)$ is a representation of a group element $g \in \mathcal{G}$. Every projective representation is characterized by the set of phases $\omega_2(g_1,g_2) = e^{i\phi(g_1,g_2)}$, known as a 2-cocycle, which are strongly constrained by associativity[18]. For a quantum system invariant under the projective representation (1), the period of the 2-cocycle (i.e the minimum number $m$ such that $[\omega_2(g_1,g_2)]^m = 1$ for all $g_1, g_2$) determines the dimension of the degenerate space. Given a nontrivial 2-cocycle, representations of at least some group elements do not commute, even for an Abelian group $\mathcal{G}$, i.e.

for some $g, h \in \mathcal{G}$, $[D(g), D(h)] \neq 0$. One can thus label the eigenstates by eigenvalues of say $D(g)$ and generate a different state, with the same energy by acting on it with $D(h)$. Notice that this argument implies degeneracy of an entire spectrum.

### Degenerate dark states in Lindbladian evolution.
The way symmetry operators act on the Lindbladian operators is rather different from the Hamiltonian case. In a system with a combined unitary and dissipative dynamics, the most general Markovian evolution of the density matrix operator, $\rho$, is described by the quantum master equation, $\dot{\rho} = \mathcal{L}(\rho)$, with

$$\mathcal{L}(\rho) = -i[H, \rho] + \sum_i \left( \ell_i \rho \ell_i^\dagger - \frac{1}{2}\left( \ell_i^\dagger \ell_i \rho + \rho \ell_i^\dagger \ell_i \right) \right). \quad (2)$$

Here $H$ is an effective Hamiltonian that represents the unitary evolution. Generically non-Hermitian, quantum jump operators, $\ell_i$, describe environment-induced dissipation effects[19,20,24].

A Lindbladian is invariant under an irreducible unitary representation $D(g)$ with $g$ an element of $\mathcal{G}$, if the Hamiltonian and the quantum jump operators satisfy[21,22]

$$D(g)HD^\dagger(g) = H; \qquad D(g)\ell_i D^\dagger(g) = \sum_j \mathcal{U}_{ij}^{(g)}\ell_j \quad (3)$$

(also known as weak symmetry[21]) where $\mathcal{U}^{(g)}$ is a unitary matrix that depends on $g$. In particular, if $\sigma$ is an eigenmatrix of $\mathcal{L}$ with an eigenvalue $\lambda_\sigma$, i.e. $\mathcal{L}(\sigma) = \lambda_\sigma\sigma$, then $D(g)\sigma D^\dagger(g)$ is an eigenmatrix with the same eigenvalue. These eigenmatrices obtained from $\sigma$ by conjugation can represent either the same, or different states. For a projective representation $D(g)$, satisfying (1), $D(g)\sigma D^\dagger(g)$ and $\sigma$ are necessarily different for some element $g$. This is because if for a particular element $h \in \mathcal{G}$, $D(h)\sigma D^\dagger(h) = \sigma$, then $\sigma$ and $D(h)$ share eigenvectors, so one can take an element $g$ such that $D(g)$ does not commute with $D(h)$. Given that $[D(g), D(h)] \neq 0$, $D(g)\sigma D^\dagger(g)$ and $\sigma$ do not share eigenvectors, meaning that they are different. By virtue of the Schur's lemma[18], the only case where this logic fails is for $\sigma$ a fully mixed state, proportional to the identity matrix.

Focusing on the case of projective representations of Abelian groups, where $D(h)D(g) = e^{i\tilde{\phi}(h,g)}D(g)D(h)$ (here $\tilde{\phi}(g_1,g_2) = \phi(g_1,g_2) - \phi(g_2,g_1)$ is again a cocycle) one can determine the dimension of the degenerate subspace in terms of the factor $e^{i\tilde{\phi}(h,g)}$. Focusing on an eigenvector $|e\rangle$ of $D(h)$ with eigenvalue $e^{i\alpha}$, the operator $D(g)$ acts on this state as a cyclic raising operator, as $D(h)D(g)|e\rangle = e^{i(\alpha + \tilde{\phi}(h,g))}D(g)|e\rangle$. For $\tilde{\phi}(h,g) = \frac{2\pi a}{m}$, with $a, m$ co-prime integers, one can raise a state $m$ times before it returns to itself. This implies that all eigenspaces of $\mathcal{L}$ are $m$-fold degenerate. In particular, the stationary subspace, defined by the eigenvalue $\lambda = 0$, is $m$-dimensional. Note that the degeneracies of the eigenspaces of $\mathcal{L}$ do not translate into degeneracies of the density matrix in general, as the eigenmatrices of $\mathcal{L}$ need not to be self-adjoint, but can come in pairs $\{\sigma, \sigma^\dagger\}$ with eigenvalues $\{\lambda_\sigma, \lambda_\sigma^*\}$. For the DDS, this is not a problem as $\lambda = 0$.

Although conditions presented above are sufficient for the existence of DDS, they are actually excessive and impractical. Indeed, the symmetry operators $D(g)$ split the entire Hilbert space into sectors of different quantum numbers that do not mix during the evolution. In other words, there is a certain number of conservation laws, which confine the long time evolution of any initial state to only a limited fraction of DDS. To access the entire DDS, this needs to be avoided. In order to achieve this, we consider states $\rho$ in the DDS that satisfy

$$\ell_i\rho = 0 \quad \text{for all } i. \quad (4)$$

We call such DDS frustration free, as $\rho$ is annihilated individually by each quantum jump operator, $\ell_i$. For systems that satisfies

Eq. (4), one can deform quantum jump operators in a way that the symmetry is broken in all the decaying subspaces, while it is maintained within the DDS. With this in mind, we define dressed quantum jump operators, that do *not* satisfy Eq. (3), as $\tilde{\ell}_i = R_i^\dagger \ell_i$, where $R_i^\dagger$ are for now arbitrary local operators. Dynamics, generated by the dressed operators, does not, in general, obey any conservation laws. Yet, $\tilde{\ell}_i$ still satisfy Eq. (4), which preserves the DDS and its degenerate multidimensional nature. For properly dressed quantum jump operators, a generic initial state evolves into a state within DDS, which has projections on all of its basis eigenmatrices.

Below we demonstrate these considerations on a 1D model, borrowing intuition from the well-studied physics of the Laughlin states on a torus[25,26]. In particular, we demonstrate that the system is driven to DDS regardless of the nature of an initial state (pure or mixed). We also devise adiabatic time-dependent Lindbladians that guarantee that the initial state is fully driven into the DDS, resulting in a state with a maximized purity.

**Laughlin states in a narrow torus geometry.** A quantum Hall droplet of $N$ electrons subject to a magnetic flux $N_\Phi = mN$ (in units of the flux quanta) and filling fraction $N/N_\Phi \equiv \nu = 1/m$ (with odd integer $m$) develops nontrivial correlations that are reproduced by Laughlin wavefunctions[27]. In a torus with periods $L_x$ and $L_y$ (with distances measured in units of the magnetic length), the area is related to the flux as $L_x L_y = 2\pi N_\Phi$. The Laughlin states at filling $\nu = 1/m$ correspond to exact zero energy states of a local Hamiltonian[28], which, after projecting onto the lowest Landau level (LLL), takes the form $\mathcal{H} = \sum_n (\ell_{0,n}^\dagger \ell_{0,n} + \ell_{1,n}^\dagger \ell_{1,n})$, where the operators $\ell_{s,n}$ ($s = 0, 1$) are[29] (see "Methods")

$$\ell_{s,n} = \sum_{l \geq 0} \eta\left(l + \frac{s}{2}\right) c_{n-l-s} c_{n+l}. \tag{5}$$

Here $c_n$ destroys an electron at orbital $n$; $\eta(l) \propto e^{-(\kappa l)^2}$ is a fast decaying function in the narrow torus limit, $\kappa^2 = \frac{2\pi}{N_\Phi} \frac{L_x}{L_y} \gg 1$ (see "Methods"). The crucial property of Laughlin states $|\Psi\rangle$ that makes them useful for our discussion of the Lindbladian evolution is that they satisfies $\ell_{s,n} |\Psi\rangle = 0$, for $s = 0, 1$ and all $n$.

In the quantum Hall context, the operators $D(g)$ of Eq. (3) correspond to inserting fluxes through the two periods of the torus (Fig. 1). In the 1D representation they are the translation operator $T$ and the operator $U$, which measures the center of mass of the particles in orbital space. They are given by

$$T = \exp\left\{\frac{2\pi i}{N_\Phi} \sum_{k=0}^{N_\Phi - 1} k \hat{\hat{n}}_k\right\}; \qquad U = \exp\left\{\frac{2\pi i}{N_\Phi} \sum_{l=1}^{N_\Phi} l \hat{n}_l\right\}, \tag{6}$$

where $\hat{n}_l$ and $\hat{\hat{n}}_k$ are the number operators at position $l$ and with momentum $k$, correspondingly. Note that $T$ and $U$ satisfy $T U T^\dagger = e^{-2\pi i \nu} U$. They thus provide a projective representation of the group $\mathcal{G} = \mathbb{Z}_m \times \mathbb{Z}_m$ with $m = 1/\nu$. This group is represented by $D(g) = U^a T^b$, where $a, b = 0, \ldots, m - 1$, while $U^m = T^m = 1$ and $UT = \omega TU$ with $\omega = \exp(2\pi i/m)$.

One can choose a basis $|\Psi_a\rangle$, such that, e.g., operator $U$ is diagonal, with its eigenvalues, $\omega^a$, along the diagonal. In this basis, the operator $T$ acts as a raising operator, because, if $U|\Psi_a\rangle = e^{\frac{i2\pi a}{m}}|\Psi_a\rangle$, then $UT|\Psi_a\rangle = e^{\frac{i2\pi(a+1)}{m}}T|\Psi_a\rangle$. In the context of the quantum Hall effect, states $|\Psi_a\rangle$ are the $m$-fold degenerate Laughlin ground states on the torus[25,30].

Note that, although the construction of the Lindbladian using the projective symmetry ensures that the eigenvalue $\lambda = 0$ of $\mathcal{L}$ is $m$-fold degenerate, the frustration-free condition Eq (4) enlarges

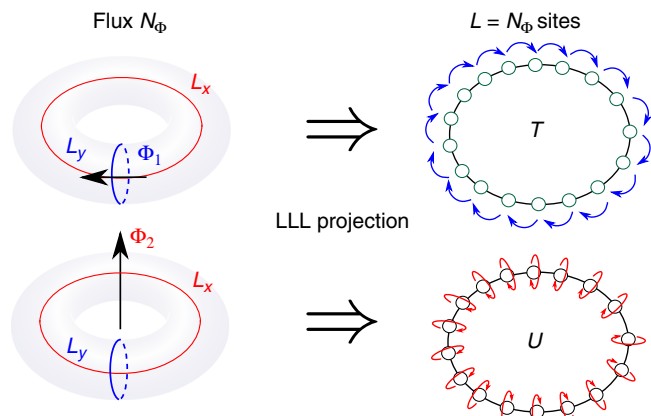

**Fig. 1 LLL projection and flux insertion in the quantum Hall liquid.** The LLL projection of a quantum Hall liquid on a torus maps the two-dimensional state into a one-dimensional ring of particles in orbital space. Inserting a flux quanta through one of the two cycles of the torus (depicted in red and blue) corresponds to a unitary operation that acts between the different ground states in the torus. In the one-dimensional representation, these unitary operations correspond to translation of the guiding centers by one orbital ($T$), or multiplication by a phase ($U$), depicted by blue and red arrows, respectively.

the degeneracy of $\lambda = 0$ to $m^2$. This is seen as follows: The symmetry operators $U$ and $T$ ensure that the density matrices $|\Psi_a\rangle\langle\Psi_a|$ for $a = 0\ldots m - 1$ share the same eigenvalue (they are related by conjugation with $T$). In general, the matrices $|\Psi_a\rangle\langle\Psi_{a+p}|$ are related, for a fixed $p$, by conjugation with $T$. Now these $m$ different families (each labeled by $p = 0\ldots m - 1$) share the same eigenvalue between them due to the frustration-free condition which ensures that if $|\Psi_a\rangle$ is annihilated by the quantum jump operators, then all the matrices $|\Psi_a\rangle\langle\Psi_b|$ are as well.

**Lindblad operators from quantum Hall physics.** In the narrow torus limit, $\kappa \gg 1$, one can truncate expressions for operators $\ell_{s,n}$ in Eq. (5), which become short-range in $n$. In this limit we have (switching from the orbital guiding center index $n$ to the real space site index $i$)

$$\ell_{0,i} = c_i c_{i+2} \quad \text{and} \quad \ell_{1,i} = c_i c_{i+1} + \beta c_{i-1} c_{i+2}, \tag{7}$$

where $\beta = \eta(\frac{3}{2})/\eta(\frac{1}{2}) = 3e^{-2\kappa^2}$. These operators transform as $U\ell_{s,j}U^\dagger = e^{\frac{4\pi i}{N_\Phi}(j+1-\frac{s}{2})}\ell_{s,j}$ and $T\ell_{s,j}T^\dagger = \ell_{s,j+1}$. Hereafter we regard $\beta$ as an arbitrary parameter. We now employ these results to construct quantum jump operators that drive the system into the frustration free DDS. In contrast with the $\ell_{s,i}$ operators in the previous section, here these operators represent processes in a real lattice of $N_\Phi$ sites, where $c_i$ destroys a fermion at the lattice site $i$. We assume $m = 3$ and the fermion density $\nu = 1/3$.

We also assume a purely dissipative evolution $\dot{\rho} = \mathcal{L}(\rho)$ with $H = 0$, Eq. (2), and the quantum jump operators $\tilde{\ell}_i = R_i^\dagger Q_i$ with $R_i^\dagger = c_i^\dagger c_{i+1}^\dagger + t(\ell_{0,i}^\dagger + \ell_{0,i+1}^\dagger)$, and

$$Q_i = \ell_{1,i} + A(\ell_{0,i-1} + \ell_{0,i+1}) + B(\ell_{1,i+1} + \ell_{1,i-1}). \tag{8}$$

Here operators $Q_i$ are linear combinations of the operators that annihilate the Laughlin states. Parameters $t$, $B$, $A$, and $β$ are determined by a realization of the dissipative dynamics and are non-universal. The DDS only depends on $β$. For the purposes of this work we take them as free parameters.

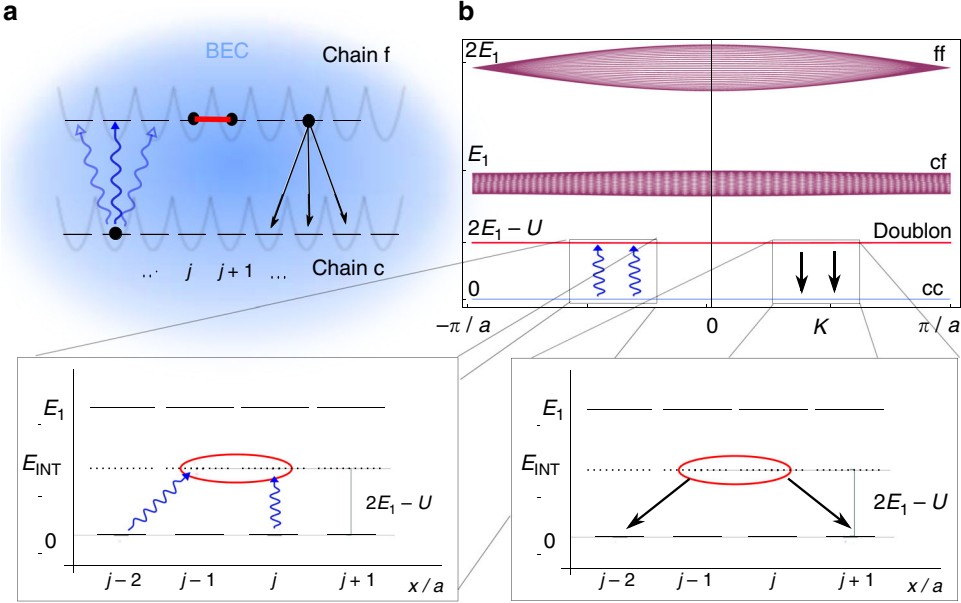

**Fig. 2 Implementation of Lindblad operators. a** Two chains c and f filled with fermions are immersed in a bath, realized as a Bose–Einstein condensate (BEC). Transitions between the two chains that have different chemical potential are mediated by the absorption of a photon from an external laser (blue wavy lines), or due to the relaxation induced by the bath (black arrows). Particles in the upper band are subject to nearest neighbor attractive interactions, of magnitude $U$ (red line). **b** Two-particle excitation spectrum, consisting of three free-particle bands (cc, cf, and ff) and the doublon band as a function of the total momentum of the pair. The laser frequency is red detuned from the transition energy $2E_1−U$, so that the laser mainly creates doublon excitations. **b** Left inset: In second-order perturbation theory in $\Omega$, the radiation couples the lower two-particle band with the doublon band. **b** Right inset: These excitations can decay back to the lower band by emitting phonons in the bath.

**Realization of Lindblad operators** $\tilde{\ell}_i$. The dissipative evolution dictated by the $\mathcal{Q}_i$ operators can be obtained by coupling two systems: a target fermion chain (where $c_i$ destroys a fermion at site $i$) with no intrinsic dynamics ($H_0 = 0$) and a fermionic chain with Hubbard interactions described by the Hamiltonian $H_1 = \sum_i J_1(f_i^\dagger f_{i+1} + \text{h.c.}) - U\sum_i n_i n_{i+1} + E_1 n_i$, where $f_i$ destroys a fermion in this at position $i$ in this chain and $n_i = f_i^\dagger f_i$. We assume that the chemical potential $E_1$ is the largest energy scale in the system. These two chains interact through an external classical radiation, with coupling Hamiltonian $H_{\text{rad}} = \Omega \cos(\omega t)\sum_i f_i^\dagger(c_i + \alpha(c_{i-1} + c_{i+1})) + \text{h.c.}$, where $\Omega$ is the intensity of the radiation, $\alpha$ parameterizes the spatial laser envelope, and $\omega$ is the frequency of the monochromatic light. The role of the driving is to excite particles from the target chain to the interacting chain. The particles then relax to the lower energy state interacting with the bath, which provides dissipation. These components are shown in Fig. 2a.

As a motivation for this construction, let us consider how a charge-density-wave (CDW) state decouples from the dynamics in this context, in the limit $J_1 << U$ and $\alpha = 0$, becoming a dark state. At first order in $\Omega$, exciting a single particle from the c to the f chain is strongly suppressed as $E_1$ is large. But in second order ($\frac{\Omega^2}{2E_1 - U - \omega}$), the radiation Hamiltonian can create two states in the f chain, which can bound in a doublon, consisting on a tightly bound pair of fermions (due to the Hubbard attraction). The wavefunction of the doublon decays exponentially with the distance $d$ between its two constituents as $t^d$ with $t \sim J_1/U \ll 1$. This means that, for the laser to create a doublon, the particles in the c chain should be near each other, as the laser acts locally. In particular, the state which locally contains nearby fermions $c_i^\dagger c_{i+1}^\dagger|0\rangle$ will be affected by radiation, as well as $c_i^\dagger c_{i+2}^\dagger|0\rangle$, where $|0\rangle$ is the state with no particles. The first local configuration that is not affected by radiation, becoming dark is $c_i^\dagger c_{i+3}^\dagger|0\rangle$, as the

fermions are too far away to be excited into a configuration with a non-vanishing matrix element with the doublon. The whole system will decouple when reaches one of the CDW states. Increasing the range of the laser (by letting $\alpha \neq 0$) creates superpositions.

The Lindblad operators (8) are obtained considering transitions between the doublon band and the low-energy band in the system (more details in the Supplementary Note 4). After performing the rotating wave approximation to account for the time dependence of the radiation Hamiltonian, the dynamics of the system occurs between the lower c and the doublon bands (see Fig 2b, upper panel). Using second-order perturbation theory in $\Omega$, we obtain the matrix elements of the transitions between the doublon states $|d_i\rangle = f_i^\dagger f_{i+1}^\dagger|0\rangle$ and the lower band states $|i,j\rangle = c_i^\dagger c_j^\dagger|0\rangle$. The transition process from lower band to doublon reads (after adiabatic elimination of the doublon) $\tilde{\ell}_i = A_\Omega^2 c_i^\dagger c_{i+1}^\dagger \mathcal{Q}_i$, valid for $t = \alpha^3 \ll 1$, with $\mathcal{Q}_i$ given in (8). The prefactor $A_\Omega \sim \Omega^2/(2E_1 - U - \omega)$, while $A$, $B$, and $\beta$ entering the definition of $\mathcal{Q}$ satisfy $A = \alpha$, $B = \alpha A_\Omega$, and $\beta = \alpha^2$. The fermion operators in $\mathcal{Q}_i$ all act on the chain c. Finally, taking into account the transitions from the doublon back to the c chain, which is mainly mediated by the dissipation with the bath, and integrating out the doublon states, we arrive at the Lindblad operators (8).

**Structure of the DDS**. Heuristically, one can understand the roles of $R_i^\dagger$ and $\mathcal{Q}_i$ as follows. Operator $\mathcal{Q}_i$ checks if at the site $i$ the state matches the local configuration of one of the Laughlin-like states. If true, it gives zero and the system stops evolving locally; if false, the operator $R_i^\dagger \mathcal{Q}_i$ scrambles the particles. As long as this process can efficiently mix the particles locally, all states in the Hilbert space eventually evolve into DDS, spanned by the three Laughlin states. Crucially, the decay into these states is a consequence of

the projective symmetry, enforcing existence of the degenerate space with $\mathcal{Q}_i\boldsymbol{\rho} = 0$ and thus $\dot{\boldsymbol{\rho}} = \mathcal{L}(\boldsymbol{\rho}) = 0$.

The basis of such DDS is formed by the Laughlin states $|\Psi_a\rangle$, where $a = 0,\ 1,\ 2$, which are annihilated by all composite operators, $\ell_{s,i}|\Psi_a\rangle = 0$, for all $s$, $i$ (and thus by the quantum jump operators). Assuming periodic boundary conditions, these states are given by[29,31] MPS[32,33]

$$|\Psi_a\rangle = \mathcal{N}\ \mathrm{tr}\left\{g_1^a g_1^a \cdots g_{L/3}^a\right\}, \tag{9}$$

where $\mathcal{N}$ is a normalization factor, $a = 0, 1, 2$ and

$$g_i^0 = \begin{pmatrix} |\circ\circ\bullet\rangle_i & |\circ\circ\circ\rangle_i \\ -\beta|\bullet\bullet\circ\rangle_i & 0 \end{pmatrix}; \quad g_i^1 = \begin{pmatrix} |\bullet\circ\circ\rangle_i & |\circ\bullet\bullet\rangle_i \\ -\beta|\circ\circ\circ\rangle_i & 0 \end{pmatrix}$$

$$g_i^2 = \begin{pmatrix} |\circ\bullet\circ\rangle_i & |\circ\circ\bullet\rangle_i \\ -\beta|\bullet\circ\circ\rangle_i & 0 \end{pmatrix} \tag{10}$$

The state $|\circ\circ\circ\rangle_i$ represents the three consecutive empty sites at positions $(3i-2,\ 3i-1,\ 3i)$, while a full dot represents an occupied site, e.g. $|\bullet\circ\circ\rangle_i = c_{3i-2}^\dagger|\circ\circ\circ\rangle_i$, etc. The dark space basis vectors, $|\Psi_a\rangle$, are related by a translation $T$ by one site, as $T|\Psi_a\rangle = |\Psi_{a\oplus 1}\rangle$, where $\oplus$ is an addition modulo 3. In the basis $|\Psi_a\rangle$, the operators $T$ and $U$ are represented by the $3 \times 3$ matrices

$$T = \begin{pmatrix} 0 & 1 & 0 \\ 0 & 0 & 1 \\ 1 & 0 & 0 \end{pmatrix}; \quad U = \begin{pmatrix} 1 & 0 & 0 \\ 0 & \omega^2 & 0 \\ 0 & 0 & \omega \end{pmatrix}; \quad \omega = e^{\frac{2\pi i}{3}}. \tag{11}$$

Within DDS the density matrix is $\boldsymbol{\rho} = \sum_{a,b=0}^{2} \varrho_{ab}|\Psi_a\rangle\langle\Psi_b|$, with $\varrho_{ab}$ a $3 \times 3$ positive semidefinite, hermitian matrix with unit trace. In general, the structure of the density matrix within DDS depends on initial conditions, which determine the parameters $\varrho_{ab}$.

The basis vectors that define the DDS depend explicitly on the parameter $\beta$. For $\beta = 0$ the DDS is spanned by a mixture of three different classical CDW[30], e.g. $|\circ\bullet\circ\circ\bullet\circ\ldots\rangle$, with periodic density and sharp structure factor in each basis vector. Changing this parameter modifies the average local density. The latter is given by

$$\mathrm{Tr}(\boldsymbol{\rho}\ \hat{n}_{3i+a}) = \frac{3\varrho_{aa} - 1}{2\sqrt{1 + 4|\beta|^2}} + \frac{1}{2}(1 - \varrho_{aa}), \tag{12}$$

indicating that, if $\beta$ is known, the local density measurements are enough to determine the probabilities $\rho_{aa}$.

An alternative way of characterizing the DDS is through the correlation function of local observables. To highlight the relation with the CDW states, we focus on the static structure factor

$$S_a(k) = \frac{3}{L}\sum_{i=1}^{L} \mathrm{Tr}(\boldsymbol{\rho}\ \hat{n}_a \hat{n}_{a+i-1})e^{ik(i-1)}, \tag{13}$$

shown in Fig. 3b. As $\beta$ increases, the system transitions from a crystal-like state with the well-defined translational symmetry of three sites into a more homogeneous state, where the density is uniform across the system.

Finally, to describe the quantum nature of the DDS basis states, we compute their entanglement entropy. We separate the degrees of freedom of the system in two complementary large regions $A$ and $A^c$ and define the partial density matrix $\rho_A = \mathrm{Tr}_{A^c}(|\Psi_a\rangle\langle\Psi_a|)$. The entanglement entropy is then $\mathcal{S}(\beta) = -\mathrm{Tr}(\rho_A \ln(\rho_A))$. The result is shown in Fig. 3c. We observe that the entanglement entropy is monotonic with $\beta$. It reaches its maximum value of $2\ln(2)$ for MPS of bond dimension 2 at $|\beta| \to \infty$.

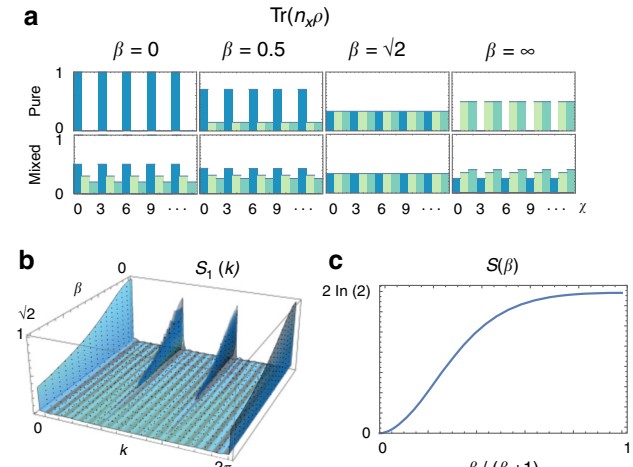

**Fig. 3 Characteristics of the DDS basis states. a** Expectation value $\mathrm{Tr}(\rho\ \hat{n}_x)$ for different positions $x$. Starting from a CDW configuration, the system is evolved using a protocol with a given $\beta$. This generates a DDS state that depends on $\beta$. The top row represents the pure state $\varrho_{00} = 1$, while the second row represents the mixed state $\varrho_{00} = 0.5$, $\varrho_{11} = 0.3$, $\varrho_{22} = 0.2$. Different colors are used to help track the changes in average occupation at each site and to highlight the 3-site periodicity of the density. **b** Static structure factor $S_1(k)$ for different values of $\beta$, for a pure state with $\rho_{11} = 1$. At $\beta = 0$ the system is in the CDW state, with a definite spatial periodicity, indicated by the peaks in $S_1(k)$ at $k = 0, \frac{2\pi}{3}, \frac{4\pi}{3}$. Increasing $|\beta|$, the system becomes more homogeneous. **c** Entanglement entropy of the DDS as a function of $\beta$.

**Time evolution and global diagnostics**. Now that we have constructed a dissipative evolution that drives the system into the DDS, we discuss how the system approaches the DDS. We analyze the Lindbladian evolution with the quantum jump operators (8) numerically. The decay into the DDS is evaluated using a quench protocol: the system is initiated in the CDW state $|\circ\bullet\circ\circ\bullet\circ\ldots\rangle$, which is one of the dark states of the Lindbladian at $\beta = 0$. This state is evolved then using the Lindblad operators (8) with $A = B = t = 1$ and $\beta \in [0, 1]$ for simplicity (the results are qualitatively similar for slightly varying these parameters). In order to obtain the evolution of the system we perform exact diagonalization using Runge–Kutta (RK) integration[34] of the master equation, for systems of sizes up to $L = 15$.

To characterize the steady-state mixture, we compute the purity of the state, defined as $\gamma(t) = \mathrm{Tr}\{\rho^2(t)\}$. From Fig. 4a, we find that the purity approaches 1/3 with larger system sizes. This is indeed the case for a sudden quench from $\beta(t \le 0) = 0$ to $\beta(t > 0) = 1$. In this scenario, the system explores an extensive portion of the Hilbert space, becoming highly mixed, as seen in the intermediate region of Fig. 4a, where the purity plateaus at a minimum. Only after the system is sufficiently mixed, it starts approaching DDS and its purity increases. The information about the initial state is practically lost in the intermediate mixing process, and the eventual steady state is a highly mixed state within DDS.

Convergence to DDS, spanned by Laughlin-like dark states $\{|\Psi_a\rangle\}_{a=0,1,2}$, may be visualized by

$$D_{\mathrm{DDS}}(t) = \mathrm{Tr}\{\rho(t)\mathcal{P}_{\mathrm{DDS}}\}, \tag{14}$$

where $\mathcal{P}_{\mathrm{DDS}} = \sum_{a=0}^{2} |\Psi_a\rangle\langle\Psi_a|$ is a projector onto the DDS. Figure 4b shows that the system indeed evolves towards the Laughlin-like DDS, proving that this is the only non-decaying

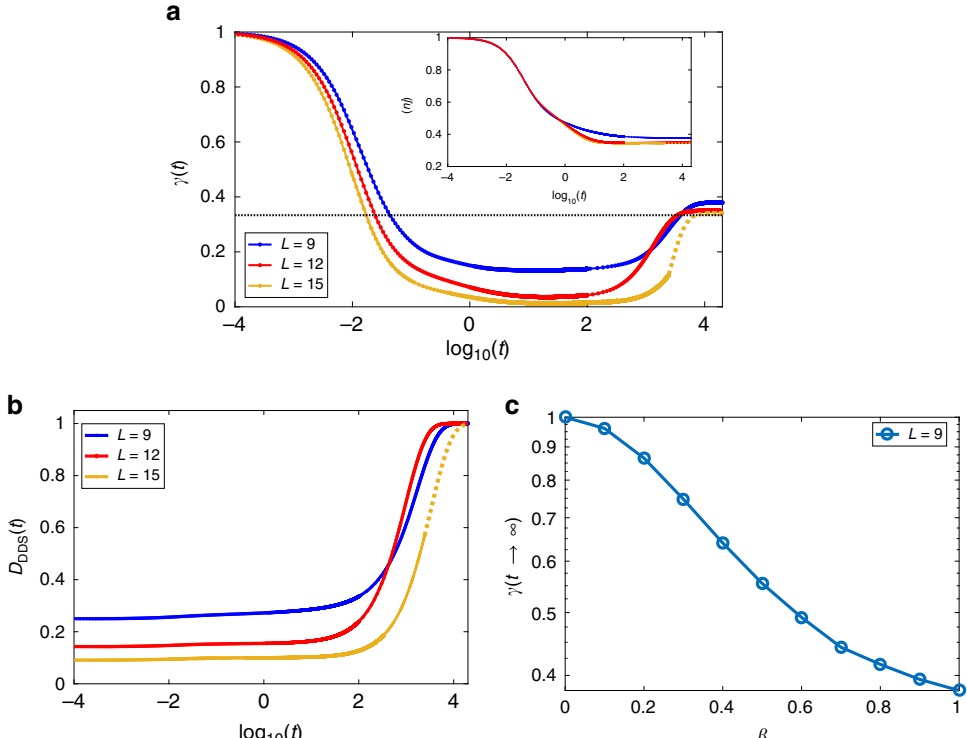

**Fig. 4 Time evolution and decay into the DDS. a** Evolution of the purity $\gamma(t)$, for different sizes. Starting from a pure CDW state, the system becomes highly mixed before starting leaking into the DDS. The dotted line shows the minimum purity possible in the DDS, corresponding to a fully mixed state. Inset: Evolution of a local observable (density). We observe that the local density relaxes to it value in the DDS in a shorter timescale compared with the time required for the system to enter the DDS. This happens independently of the system size. **b** Approach of the DDS, spanned by the Laughlin-like states on the narrow torus, for different system lengths $L$. **c** Purity in the DDS as a function of $\beta$. The purity of the DDS depends strongly on $\beta$, indicating that the purity can be maintained if the final state is close to the initial one. The dotted lines for the $L = 15$ case for the purity and density evolution are obtained from an approximation for the Lindbladian on a smaller subspace. In the approximation we neglect the terms of the evolved state $\rho(t)$ which are smaller than a threshold $\epsilon = 10^{-5}$ and construct the effective Lindbladian for the remaining subspace, which has a smaller dimension. The dotted lines for $L = 15$ for the $D_{\mathrm{DDS}}$ are obtained from the knowledge of the ADR, and simply performing a continuation of the dynamics.

subspace of the Lindbladian evolution. At large times, one finds $1 - D_{\mathrm{DDS}}(t) \propto e^{-\lambda_0 t}$, where the rate $\lambda_0$ is given by the lowest non-zero eigenvalue of the Lindblad operator. (A longer discussion regarding this gap for finite systems is developed in the supplementary note 1.)

One notices that $\lambda_0$ is slowly decreasing upon increasing the system size. We note that local observables, like the particle density, fast approach their steady-state values in a way which is independent of the system size (Fig. 4a). This separation of scales indicates that, while locally the system reaches a configuration that is close to the dark states that span the DDS, globally it takes much longer to fully reach the DDS.

If instead of quenching the system into $\beta = 1$, we quench it into $\beta \ll 1$, we observe a very different behavior. Here the system does not have to explore an extensive part of the Hilbert space before it reaches the DDS. As a result, the purity remains close to 1 at all times, as can be seen in Fig. 4c.

**Adiabatic evolution.** Although the previous analysis shows that the system does not generically end up in a pure state, it is possible to increase the purity of the final state by performing an adiabatic evolution from a pure state[35–37]. To illustrate this, we evolve the system from an initial state given by a superposition of the three CDW configurations. This allows us to characterize the coherences in the MPS basis throughout the adiabatic evolution (Sec. III in SI). Individual CDW states can be created using

existing experimental techniques[38]. We then evolve this state with the Lindblad operators (8) using a time-dependent $\beta$ parameter: $\beta(t) = \Delta \cdot t$ for $0 < t \le 1/\Delta$ and $\beta(t) = 1$ for $t > 1/\Delta$, where $\Delta$ is the ramp velocity. The purity of the final state depends on $\Delta$ as shown in Fig. 5a. For small enough $\Delta$, the system does not explore the whole Hilbert space, but instead remains almost pure throughout its entire evolution. This mechanism can be used to achieve a purity arbitrarily close to unity. For larger ramp velocities, the system rapidly departs from the initial state, exploring the many-body Hilbert space, as shown for intermediate times in Fig. 5b, before leaking back to the DDS, which remains the only attractor of the dynamics. This increases the departure of the steady state from a pure state and erases the information about the initial state. The steady-state purity as a function of the ramp velocity is shown in Fig. 5c.

## Discussion

We have shown that to achieve a DDS the Lindbladian evolution should have an underlying symmetry, admitting a projective representation. The period of its 2-cocycle determines the dimensionality of the dark space. Reaching a DDS, protected against environmental influence, offers a way of maintaining quantum information. To manipulate this information, it is necessary to have an access to high-purity states within the DDS. We found that an adiabatically varying Lindblad operator allows to reach such nearly pure, entangled states. We have

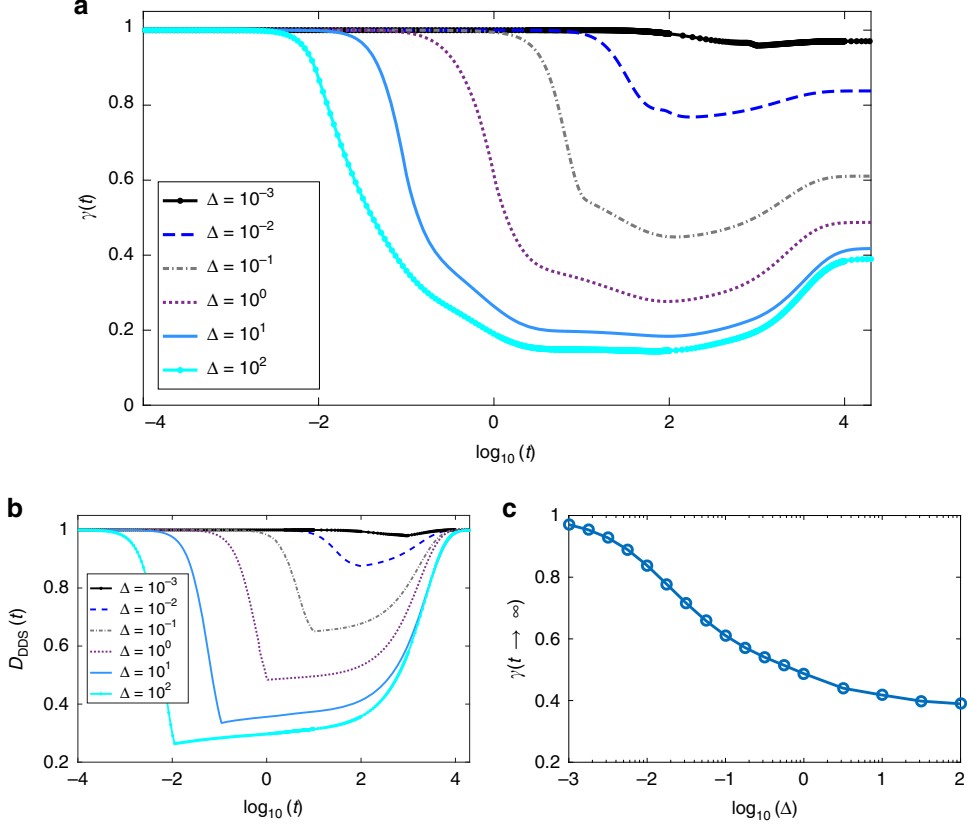

**Fig. 5 Adiabatic evolution of the system. a** The purity of the final state for different ramp velocities $\Delta$ in a system with $L = 9$ sites. A slower variation in the adiabatic protocol leads to a higher purity. **b** The system ends up in the DDS regardless of the ramp velocity. Different $\Delta$'s control how much of the Hilbert space is explored. **c** The purity of the final state can be manipulated via the ramp velocity.

demonstrated these ideas by studying the thin torus limit of the $v = 1/3$ fractional quantum Hall state of matter. Being able to generate and manipulate states within a DDS may be utilized for quantum information processing platforms. The many-body nature of the state renders it less fragile against local disturbances.

## Methods

**Mapping fractional quantum Hall ground state to one-dimensional model.** In this section we revisit the exact mapping of the Laughlin state[27] into a one-dimensional state[39,40]. We will be interested in filling fractions $v < 1$. Recalling that a 2DEG in a strong magnetic field displays Landau levels, we assume that the relevant physics occurs in the LLL. We place the system into a 2D torus with linear sizes $L_x$ and $L_y$ and area $A = L_x L_y \sin \theta$, defined by the region in the upper half complex plane enclosed by the points $w = (0, L_x, L_y \tau, L_x + L_y \tau)$. This torus is characterized by the modular parameter $\tau = L_y/L_x e^{i\theta} = \tau_1 + i\tau_2$, $(\text{Im}(\tau) > 0, \theta \in [0, \pi])$. Following ref. [25] we introduce the translation operators $t(\mathbf{L}) = \exp(\mathbf{L} \cdot (\nabla - ie\mathbf{A}) - iL_x y + iL_y x)$ (here $\mathbf{L} = (L_x, L_y)$ are measured in units of the magnetic length) which correspond to the usual translation operators in terms of the canonical momentum, and an extra space-dependent phase. The single-particle wavefunction satisfies the boundary conditions $t(\mathbf{L_a})\Psi = e^{i\phi_a}\Psi$, with $\mathbf{L_a}$ a translation over the lattice vectors $\mathbf{L_1} = (L_x, 0)$ and $\mathbf{L_2} = L_y(\cos\theta, \sin\theta)$. Both conditions can be satisfied if the flux over the torus $\frac{L_x L_y \sin\theta}{2\pi} = N_\Phi$ is integral. We parameterize the coordinates on the torus by $z = \tilde{z}/L_x$ with $\tilde{z} = L_x(x + y\tau)$, where $x \in [0, 1]$ and $y \in [0, 1]$.

The relation with the usual Cartesian coordinates is $x_1 = L_x(x + \tau_1 y)$ and $x_2 = L_y \tau_2 y$. The single-particle wavefunction has the form $\Psi = e^{-\frac{1}{2}(\text{Im}(\tau)L_x y)^2} f(z)$, where $f(z)$ is an entire (holomorphic) function in the complex plane. Then we use units where $\sqrt{\hbar/eB} = 1$. In the Landau gauge $\mathbf{A} = -By\hat{x}$, the boundary conditions read

$$f(z + 1) = f(z)e^{i\phi_1},$$
$$f(z + \tau) = f(z)e^{i\phi_2}e^{-i\pi N_\Phi(2z + \tau)}, \tag{15}$$

where the phases $\phi_a$ correspond to the fluxes piercing the torus in the two orthogonal directions $a = 1, 2$. From these relations it follows that $\int \mathrm{d}z \frac{\mathrm{d}}{\mathrm{d}z} \ln(f(z)) = 2\pi i N_\Phi$, which implies that the function $f(z)$ has $N_\Phi$ zeroes. The single-particle wavefunction that satisfies boundary conditions (15) and has

$N_\Phi$ zeroes is given by the generalized theta function

$$\phi_n(z; \tau, \phi_1, \phi_2) = e^{-\frac{1}{2}(\text{Im}(\tau)L_x y)^2} \vartheta\left(z - z_n \left| \frac{\tau}{N_\Phi}\right.\right) e^{i\phi_1(z - z_n)}$$

$$\text{with} \quad \vartheta(z|\tau) = \sum_{m=-\infty}^{\infty} \left(e^{i\pi\tau}\right)^{m^2} e^{2\pi i m z} \tag{16}$$

and $z_n = \frac{2\pi n + \phi_2 - \tau\phi_1}{2\pi N_s}$. This corresponds to a normalizable wavefunction for $\text{Im}(\tau) > 0$. The zeroes of $\varphi_n(z; \tau, \phi_1, \phi_2)$ are located at $z = z_n + \frac{1}{2} + m + \left(\frac{1}{2} + n\right)\frac{\tau}{N_\Phi}$.

As shown in ref. [28], the Laughlin state at filling $v = 1/3$ is the zero energy exact ground state of the Landau problem with the interaction $\mathcal{H} = V_0 \int \mathrm{d}\mathbf{r} |\nabla \rho(\mathbf{r})|^2$, where $\rho(\mathbf{r}) = \psi^\dagger(\mathbf{r})\psi(\mathbf{r})$ and $\mathbf{r} = (x_1, x_2)$. The projection of the electron operator into the first Landau level is $\psi = \sum_n \varphi_n(\mathbf{r})c_n$, where $c_n$ destroys a state at occupation $n$. The interaction Hamiltonian projected onto the first Landau level becomes

$$\mathcal{H} = \frac{32N_\Phi V_0}{|\tau|^3 L_1^2} \sum_j Q_j^\dagger Q_j$$

$$\text{with} \quad Q_j^\dagger = \sum_j^{N_\Phi} \sum_{k=-\infty}^{\infty} (j - N_\Phi k)e^{-\frac{2\pi i}{N_\Phi \tau}(j - kN_\Phi)^2} c_{j+k}^\dagger c_{j-k}^\dagger. \tag{17}$$

In this sum the pair of numbers $(j, k)$ are all integers or all half integers and satisfy $0 < j < N_\Phi$, $0 < k, l, < N_\Phi/2$. Separating both cases, and defining $\kappa^2 = \frac{2\pi}{N_\Phi}\frac{L_x}{L_y}$ gives the operators $\ell_{s,n}$ (Eq. 5) in the main text.

**Physical realization.** We consider a one-dimensional optical lattice (system) immersed in condensate that acts as a bath to the system, providing dissipation. Each site in the optical lattice consists of a potential-well accommodating two single-particle levels denoted by $c$ and $f$

$$H_{\text{sys}} = H_0 - U \sum_i n_{i,1} n_{i+1,1} + \sum_{i,\sigma} E_\sigma n_{i\sigma}. \tag{18}$$

Here $-\sum_{i,\sigma}^N (J_\sigma a_{i,\sigma}^\dagger a_{i+1,\sigma} + \text{h.c.})$ and $a_{i,\sigma}$ is a fermionic annihilation operator at site $i = 1 \ldots N_\Phi$ and level $\sigma = \{0, 1\}$. The relation with the main text operators is $a_{i,0} = c_i$ and $a_{i,1} = f_i$. The Hamiltonian includes an attractive ($U > 0$) interaction between neighbor particles in the level $\sigma = 1$. The operator $n_{i\sigma} = a_{i\sigma}^\dagger a_{i\sigma}$ measures the

occupation at the site $i$ and level $\sigma$. We assume that the number of particles in the system is given by $N_e = N_\Phi/3$.

To capture the essential physics generated by the interaction, we first study the two-particle problem. Defining the two-particle state $|\sigma\sigma'\rangle = \sum_{i,j} \chi_{ij}^{\sigma\sigma'} a_{i,\sigma}^\dagger a_{j,\sigma'}^\dagger |0\rangle$, with $|0\rangle$ the state with no particles (vacuum), the Schrödinger equation for the wavefunction $\chi_{ij} = \frac{1}{\sqrt{2}}(\chi_{ij}^{11} - \chi_{ji}^{11})$ reads $-J_1(\Delta_i + \Delta_j)\chi_{ij} - U(\delta_{i,j+1} + \delta_{i+1,j})\chi_{ij} = (E - 2E_1 + 4J_1)\chi_{ij}$, with $\Delta_i \chi_{ij} = \chi_{i+1,j} - 2\chi_{ij} + \chi_{i-1,j}$, the discrete Laplace operator. Introducing the central and relative coordinates $R = a(j+l)/2$ and $r = a(j-l)$ the wavefunction can be written as $\chi_{jl} = e^{iRK}\chi(K)_r = e^{iRK}\sum_q e^{irq}\tilde{\chi}_q^s$ where we have introduced the total and relative momentum $K = k_1 + k_2$ and $2q = k_1 - k_2$. The Schrödinger equation for $\chi_{ij}^s$ becomes

$$\tilde{\chi}_q^s = \frac{2}{N_\Phi} \frac{U \sin q \, \bar{\chi}}{E - 2E_1 + 4J_1 \cos \frac{K}{2} \cos q}, \tag{19}$$

where $\bar{\chi} = \sum_q 2 \sin q \, \tilde{\chi}_q^s$. For fixed center of mass momentum, the bound state energy $E_d(K)$ is found by solving self-consistently Eq. (19), leading to $E_d(K) = 2E_1 - U - 4\frac{J_1^2}{U}\cos^2\frac{K}{2}$. We consider the regime $J_0 \sim 0$, along with $|U| \gg E_1$ and $\frac{J_1}{E_1} \ll \frac{J_1}{|U|} \ll 1$. In this case, the bound state energy $E_d(k \sim 0)$ is far below the bottom of the (1, 0)-pair band, but still above the (0, 0) two-particle band. The amplitude for tunneling between two 0 states is taken to be negligible compared to all other energy scales ($J_0 \sim 0$). This implies a flat band for the (0, 0) pair. The two-particle energies $E_{\sigma\sigma'}$ of the continuous Bloch bands are

$$E_{\sigma\sigma'}(K,q) = (\sigma + \sigma')E_1 - 2(J_\sigma + J_{\sigma'})\cos\frac{Ka}{2}\cos qa$$
$$+ 2(J_\sigma - J_{\sigma'})\sin\frac{Ka}{2}\sin qa, \tag{20}$$

A doublon state of definite momentum is created by the combination $d_K^\dagger = \sum_R e^{iKR}\sum_\ell e^{-\ell/\xi(K)} f_{R+\frac{\ell}{2}}^\dagger f_{R-\frac{\ell}{2}}^\dagger$, with $\xi^{-1}(K) = \ln\left(\frac{J_1}{|U|}\cos\frac{K}{2}\right)$. The state $|d_K\rangle \equiv d_K^\dagger|0\rangle$ is normalized as $\langle d_{K'}|d_K\rangle = \delta_{K'K}$, with $|0\rangle$ a state with no particles.

## Data availability
The data used to create the plots are available from the authors upon reasonable request. The figures were produced using Python and processed using Inkscape.

## Code availability
The numerical codes and scripts to obtain the data are available from the authors upon reasonable request.

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

## Acknowledgements
We are indebted to R. Fazio for valuable discussions. R.A.S. acknowledges funding from EPSRC grant EP/M02444X/1, and the ERC Starting Grant No. 678795 TopInSy. F.I. acknowledges the financial support of the Brazilian funding agencies National Council for Scientific and Technological Development—CNPq (Grant No. 308205/2019-7) and FAPERJ (Grant No. E-26/211.318/2019). A.K. was supported by NSF grants DMR-1608238 and DMR-2037654. Y.G. was supported by the Deutsche Forschungsgemeinschaft (DFG) TRR 183 (project B02), and EG 96/13-1, by the Israel Science Foundation and the US-Israel Binational Science Foundation (BSF), Jerusalem, Israel.

## Author contributions
R.A.S. identified the connection between projective symmetry and dark state degeneracy and performed the analytical calculations. F.I. performed the numerical calculations. R.A.S., A.K., and Y.G. defined the problem, discussed and analyzed the physical consequences, and developed the experimental proposal for the realization of the model All authors contributed significantly to the writing of the manuscript.

## Competing interests
The authors declare no competing interests.
