## [Peer Review File · Nature Communications]

Reviewers' comments:

Reviewer #1 (Remarks to the Author):

In this paper the authors investigate how symmetries can lead to multidimensional dark states that are decoupled from the environment. They propose these dark states as a way to protect quantum information from the influence of the environment. The authors give generic technical conditions for dark spaces to exist, study their properties, and then investigate a specific example to show how a system may be driven into a dark space. The proposed model is based on fractional quantum Hall states and the resulting states are analyzed in detail. The authors do not discuss how such a model may be realized experimentally and whether current experimental technologies would allow for a sufficiently accurate implementation of the discussed model. I feel that this model is rather involved and do not see how this could be achieved. The authors also do not discuss the achievable coherence times for quantum information stored in this way when including likely experimental imperfections. A major question with any protected quantum information storage protocol is how to read, write, and control it via external fields. Unless this is viable with reasonable experimental effort the benefits of the protected storage might be lost. Furthermore, questions of scalability and whether it will be possible to perform entangling gates between qubits stored in different systems are essential for any viable quantum information processing platform. Such questions are not addressed in the publication at all. Instead the manuscript focuses on a highly technical and detailed description of the theoretical framework. Throughout the paper the relevance and importance of the findings presented in the paper remain unclear. The paper does not fulfil its promise in the title of providing a viable path towards (useful) dissipation-protected qubits (which is iterated again in the introduction). I find that this paper is thus much more appropriate for a specialized journal and feel that it is not suitable for a general audience. I can therefore not recommend this manuscript for publication in Nature Communications.

In addition to the points mentioned above, I do find that the authors have ignored what I would consider highly relevant publications on the topic of decoherence free subspaces. These include e.g. D.A. Lidar, et al., Phys. Rev. Lett. 81, 2594–2597 (1998) and E. Knill, et al., Phys. Rev. Lett. 84, 2525–2528 (2000). While these previous papers are probably less general than the generic framework described in the current paper they have still attracted a huge amount of attention and have defined the field of decoherence free subspaces for quantum information processing. In particular, after the authors make their assumption Eq. (4) their DDS seem to be identical to a well-known decoherence-free subspace. Even the more generic case has been studied and is closely related to the framework described in B. Buča et al. Nature Communications 10, 1730 (2019).

Finally, it is unclear how the framework presented in the current manuscript could be exploited to identify and engineer novel experimental platforms for dissipation-protected storage and manipulation of quantum information. The presented example is constructed to theoretically satisfy

the required conditions. However, the broader applicability of the presented framework is not discussed. I am therefore not convinced by the manuscript that this proposal will be of broad interest and relevant beyond the specific example given in the main text.

In summary, I can thus not recommend publication of this manuscript in Nature Communications.

Reviewer #2 (Remarks to the Author):

The authors propose a general route to generating degenerate dark states using the math of projective representations applied to Lindblad dynamics. They start by generally relating degeneracy to the period of the 2-cycle (phases in the projective representation). Then they argue that degeneracy of the steady state can be achieved more simply by requiring that in the steady state space, all density matrices are annihilated by the Lindblad operators, which they call frustration free. They then proceed to use this second construction to engineer a dissipative model that stabilizes Laughlin states in the thin-torus limit, propose a method to realize it through fermions coupled to a BEC, and demonstrate by quenches and adiabatic ramps to prepare the degenerate dark states (DDS).

In general I find this to be high-quality, novel work worthy of publication in Nature Communications. To the best of my knowledge, this is the first generic route to creating degenerate dark state manifolds, which could be useful for quantum. The result that pure dark states may be created adiabatically is useful as well. From a standpoint of readability, it is generally pretty well written, though there are a few parts I will mention which go too quickly or are light on details.

I suggest to publish in Nature Communications after the following suggestions/questions have been considered:

- The particular model considered of specially constructed dissipation is very hard to follow and not so clearly physically motivated. From what I can tell, the choice of model is partially motivated by the desire to stabilize Laughlin states, but the detailed terms actually come from the specific fermion+BEC realization. I cannot follow the experimental realization at all – much more detail needs to be specified to clarify the origin of the model, which currently seems to emerge almost magically from Fig. 2. I suggest bring some of the text from the Methods into the main text, as this at least motivates the model a bit better. More explanation would be great if space permits.
- I don't understand the notation $DD(gg) = UUaa \otimes TTbb$. It seems UU and TT are matrices in the full Hilbert space HH , so their product would give an object living in HH^2 , rather than a matrix in HH as I think we want for this representation.
- I find the connection between the general construction using projective reps and the explicit construction for Laughlin states to be hard to follow. It seems like the notation is chosen such that

QQ_{ii} plays the role of the original Lindblad operators ℓ_{ii} , which are scrambled by RR_{ii} to avoid breaking up into symmetry sectors. Can it be shown that the role of group representation TT and UU is to unitarily mix the Lindblad operators QQ_{ii} as in Eq. 3, such that if you just used the QQ_{ii} 's you would have a proper projective rep enforcing degeneracy, as in Eq. 2 and 3?

Reviewer #3 (Remarks to the Author):

The manuscript by Santos and coworkers presents a theory proposal that uses engineered dissipation to generate multi-dimensional degenerate dark space (DDS) to protect encoded quantum information. They build a model protocol that can stabilize DDS basis isomorphic to the degenerate Laughlin states. I have some questions that need clarification in order to decide the significance of this work:

1. What is the scaling of the dissipative gap λ_0 with system size? The authors vaguely state that " λ_0 is slowly upon decreasing with the system size" and "globally it takes much longer to fully reach the DDS", which look contradictory. It is important to clarify whether λ_0 decreases with system size in a power-law or exponential scaling.
2. It seems that the authors only tested purity for initial states associated with with three charge density wave configurations. Will the adiabatic scheme preserve purity for superpositions of the charge density wave configurations?
3. It is unclear what kind of noise will be protected by the DDS. Will it protect against local decay or dephasing errors?
4. How to process the encoded information in the DDS? If we have to switch back and forward between charge density wave basis and topological encoding, we will not benefit from topological protection when processing quantum information.

Response to referees.

Re: **Resubmission of our Manuscript NCOMMS-20-01363-T**

Multidimensional dark space and its underlying symmetries: towards dissipation-protected qubits

Reply to Referee 1

We thank the referee for her/his comments.

Referee comment

- *The authors do not discuss how such a model may be realized experimentally and whether current experimental technologies would allow for a sufficiently accurate implementation of the discussed model. I feel that this model is rather involved and do not see how this could be achieved. The authors also do not discuss the achievable coherence times for quantum information stored in this way when including likely experimental imperfections. A major question with any protected quantum information storage protocol is how to read, write, and control it via external fields. Unless this is viable with reasonable experimental effort the benefits of the protected storage might be lost. Furthermore, questions of scalability and whether it will be possible to perform entangling gates between qubits stored in different systems are essential for any viable quantum information processing platform. Such questions are not addressed in the publication at all. Instead the manuscript focuses on a highly technical and detailed description of the theoretical framework. Throughout the paper the relevance and importance of the findings presented in the paper remain unclear. The paper does not fulfil its promise in the title of providing a viable path towards (useful) dissipation-protected qubits (which is iterated again in the introduction).*

Response

Our original manuscript included a discussion of an experimental realization with cold atoms in a 1D optical lattice setup (cf. the extended caption of Fig.2 and supplemental information (SI) sections V and VI). In the new version we have significantly expanded the discussion about experimental implementation adding it in the main text, section “Realization of Lindblad operators”. Here we discuss how the use of a Hubbard-like interaction, an external driving laser of intensity Ω , and a system/bath coupling of strength g can be used to create the desired Lindblad operators, characterized by the rate $\Omega g/\Delta$, where Δ is the detuning of the laser frequency from the doublon mode energy created by the Hubbard interaction. We have included an analysis on the effects of imperfections in the Lindblad protocol, in section II of the supplemental information. There we find that for imperfections occurring at a rate ϵ , small compared with $\Omega g/\Delta$, perturbation theory reveals that there exists a window of operation given where the physics discussed in this work applies. In a broader context, while we do agree with the referee that the issues of scalability, coherence time and quantum information processing are not extensively and deeply studied in the manuscript, we feel that such a critique is unfair. One cannot expect a single publication to address all these major issues at once. Our focus here is indeed “*on a detailed description of the theoretical framework*”, providing generic principles behind the existence of the degenerate dark spaces. A specific non-trivial many-body realization serves an illustrative purpose, rather than being a fully developed

experimental proposal for a concrete device. We thus disagree respectfully with the assertion that our paper does not fulfil its promise.

Referee comment

- *In addition to the points mentioned above, I do find that the authors have ignored what I would consider highly relevant publications on the topic of decoherence free subspaces. These include e.g. D.A. Lidar, et al., Phys. Rev. Lett. 81, 2594–2597 (1998) and E. Knill, et al., Phys. Rev. Lett. 84, 2525–2528 (2000). While these previous papers a probably less general than the generic framework described in the current paper they have still attracted a huge amount of attention and have defined the field of decoherence free subspaces for quantum information processing.*

Response

We completely agree with the referee that these papers are historically important and should have been mentioned. We have corrected this omission in the new version. We note that the work of Lidar et al. indeed refers to a “decoherence free subspace”, but does not discuss the degree of degeneracy of such a subspace, nor the relation of degeneracy to underlying symmetries, optimization of purity and fidelity, or detailed many-body implementations. The same goes for Knill et al.

Referee comment

- *In particular, after the authors make their assumption Eq. (4) their DDS seem to be identical to a well-known decoherence-free subspace. Even the more generic case has been studied and is closely related to the framework described in B. Buca et al. Nature Communications 10, 1730 (2019).*

Response

This is an interesting paper, which considers a rather different problem of having an oscillatory, non-stationary long-time dynamics. The authors of that paper have found that this is the case if there is an operator commuting with all jump operators. They do not specify the symmetry condition, enforcing existence of such operator(s). Degeneracy in their case is incidental, and is not a crucial theme of their analysis. We feel that the fact that paper was accepted to Nature Communication makes a strong case for acceptance of our manuscript, with novel features concerning degeneracy, symmetry, optimized steering, and many-body implementations. We stress that engineering a degenerate dark space is what enables quantum manipulations within a protected subspace.

Referee comment

- *Finally, it is unclear how the framework presented in the current manuscript could be exploited to identify and engineer novel experimental platforms for dissipation-protected storage and manipulation of quantum information. The presented example is constructed to theoretically satisfy the required conditions. However, the broader applicability or the presented framework is not discussed. I am therefore not convinced by the manuscript that*

this proposal will be of broad interest and relevant beyond the specific example given in the main text.

Response

Although it is true that we do not present an engineering proposal for a quantum computation device, we do address the possibility of building this type of dynamics, manipulating particles on an optical lattice. A full comprehensible discussion of the limits of this experimental proposal is not the goal of our work. Instead, we uncover a generic symmetry-based framework, which must underline any dark space-based platform—the protected subspace for our quantum manipulations. We demonstrate how to apply these principles in a specific scenario involving a non-trivial many-body platform. Moreover, we demonstrate how to implement optimal steering protocols (maximizing purity and fidelity). These general and generic guidelines will be invaluable in the development of concrete working realizations. Putting our work in this context clearly underlines its importance and general interest. Presently we are working on developing some of these ideas further, but it is entirely possible that other groups will come up with utterly different (and possibly better) proposals.

Reply to Referee 2

We thank the second referee for her/his comments, in particular for their remarks “*I find this to be high-quality, novel work worthy of publication in Nature Communications. To the best of my knowledge, this is the first generic route to creating degenerate dark state manifolds, which could be useful for quantum*” and “*The result that pure dark states may be created adiabatically is useful as well.*” which acknowledge the novelty of our work.

Below we reply to the comments/questions of this referee.

Referee comment

- *The particular model considered of specially constructed dissipation is very hard to follow and not so clearly physically motivated. From what I can tell, the choice of model is partially motivated by the desire to stabilize Laughlin states, but the detailed terms actually come from the specific fermion+BEC realization. I cannot follow the experimental realization at all – much more detail needs to be specified to clarify the origin of the model, which currently seems to emerge almost magically from Fig. 2. I suggest bring some of the text from the Methods into the main text, as this at least motivates the model a bit better. More explanation would be great if space permits.*

Response

In order to demonstrate the viability of our approach, we have addressed here a non-trivial multiply-degenerate many-body state, specifically a Laughlin state. While this represents a correlated fermionic state in two dimensions, one can instead resort to a correlated state in one-dimension, akin to the Laughlin state. Once this is established, the 1D state can be mimicked by a drive and

dissipation protocol, amenable to realizations on various platforms, e.g. cold atoms. Needless to say, the principles outlined here apply to other examples of engineering degenerate many-body states.

As suggested by the referee, we have moved the methods section to the main text. We have expanded the discussion on the experimental realization to clarify its connection with the proposed model. In the main text, we have opted to highlight how the charge density wave (CDW) states can be created, deferring the details of the full realization to the supplemental material. The main point is that the energy difference between the doublon band and the target band can be provided by the laser light, so that the laser mainly creates doublon excitations (we assume that the matrix element for this process does not vanish). In the limit of weak hopping, and on-site laser excitations, the only states that decouple from the dynamics are the ones where the particles in the band lowest energy band are separated by more than 2 sites, which, for 1/3 density correspond to the CDW states.

Referee comment

- *I don't understand the notation $D(g)=U^a \otimes T^b$. It seems U and T are matrices in the full Hilbert space H , so their product would give an object living in H^2 , rather than a matrix in H as I think we want for this representation.*

Response

We thank the referee for spotting this typo. As correctly suggested, the tensor product has been replaced by a normal product in the text, i.e. $D(g)=U^a T^b$ which acts in the full Hilbert space H .

Referee comment

- *I find the connection between the general construction using projective reps and the explicit construction for Laughlin states to be hard to follow. It seems like the notation is chosen such that Q_i plays the role of the original Lindblad operators ℓ_i which are scrambled by R_i to avoid breaking up into symmetry sectors. Can it be shown that the role of group representation T and U is to unitarily mix the Lindblad operators Q_i as in Eq. 3, such that if you just used the Q_i 's you would have a proper projective rep enforcing degeneracy, as in Eq. 2 and 3?*

Response

To clarify this point, we have renamed the operators Q_i as ℓ_i as they indeed play the role of Lindblad operators in our construction. For the sake of completeness we now describe how they transform under the group representations T and U .

Reply to Referee 3

We thank the referee for her/his comments.

Referee comment

- 1. What is the scaling of the dissipative gap λ_0 with system size? The authors vaguely state that " λ_0 is slowly upon decreasing with the system size" and "globally it takes much longer to fully reach the DDS", which look contradictory. It is important to clarify whether λ_0 decreases with system size in a power-law or exponential scaling.

Response

This is indeed a very important point. To address this more reliably, we have now improved our numerical results to shed light on this question and have included an expanded analysis in the Supplementary Information (Section “Lindbladian gap in finite system sizes”). We have studied the Lindbladian gap in two different forms: (i) directly by exact diagonalization of the Lindbladian superoperator, and (ii) indirectly by the asymptotic decay rate (ADR) of the quantum state dynamics. While exact diagonalization allowed us to study the Lindbladian gap for sizes up to $L \sim 12$, the asymptotic decay rate analysis allows the study of larger $L \sim 15$ system sizes.

We have observed that the gap from exact diagonalization matches the one obtained from ADR. Furthermore, we have obtained that, within the system’s sizes we were able to analyse, the dissipative gap did not show a clear tendency towards shrinking as the length increases. Although instructive, these results for small systems do not allow us to unequivocally determine the nature of the dissipative gap as the system size increases. Nonetheless, it suggests that the gap does not close exponentially fast; such a behaviour would be detrimental for the preparation and manipulation of DDS in quantum information tasks.

Referee comment

- 2. It seems that the authors only tested purity for initial states associated with three charge density wave configurations. Will the adiabatic scheme preserve purity for superpositions of the charge density wave configurations?

Response

We thank the referee for this suggestion. We have performed this analysis, and we have updated Fig. 5 and found that the adiabatic scheme indeed preserves the purity of an initial state composed of a superposition of charge density wave configurations in a manner that is qualitatively similar to the evolution of a single charge density wave (CDW). We have also included an analysis of the coherences in time in the Supplementary Information – Sec. “Adiabatic evolution” for details. We are able to determine that for a sufficiently slow ramp, the adiabatic scheme generates, from each CDW component of the superposition, the corresponding Laughlin-like state, thus allowing to generate superpositions of topologically encoded states.

Referee comment

- 3. It is unclear what kind of noise will be protected by the DDS. Will it protect against local decay or dephasing errors?

Response

We have analysed four types of errors.

- 1.- Additional set of dephasing dissipative channels
- 2.- Imperfections in the current Lindblad operators of the form of extra hopping terms
- 3.- Coherent Hamiltonian competing with the dissipative dynamics
- 4.- Additional set of decay dissipative processes, leading to a non conservation of the total number of fermions.

Our results for these imperfections are shown in the Supplementary Material – Sec. “Lindbladian perturbations”. There we add the following discussion:

Our results for the first three imperfections above are shown in Fig.(S2). We see that perturbation theory provides a qualitative picture: in the regime of small perturbations (in units of the original Lindbladian, $\varepsilon \ll 1$), whereas imperfections in the jump operators lead to a linear splitting of the DDS, $\lambda_2 \sim \varepsilon$, a Hamiltonian perturbation leads to a quadratic dependence $\lambda_2 \sim \varepsilon^2$. Thus, as long as the perturbation is small compared to the unperturbed gap, there is a time window between the system entering the DDS and the system characteristics of the state being destroyed by the imperfections, which gives a possibility to effectively use these states for quantum information tasks.

The case of an additional set of decay dissipative channels follows in a similar form. In this case the steady state of the evolution is the vacuum state for $\varepsilon_{\text{decay}} > 0$. However, as above, if the perturbation is small there is a time window over which the effects on the DDS are negligible. One may obtain the characteristic time of the dissipative decay effects by the dynamics of the total number of particles N in the system, which in the Heisenberg picture is described by $L^\dagger[N] = -\varepsilon_{\text{decay}}N$, i.e., $N(t) \sim \exp(-\varepsilon_{\text{decay}} t)$. Thus the effects of particle losses in the system are relevant for times of the order $t \sim 1/\varepsilon_{\text{decay}}$, similarly to the other imperfections in the quantum jump operators considered above.

Referee comment

- 4. How to process the encoded information in the DDS? If we have to switch back and forward between charge density wave basis and topological encoding, we will not benefit from topological protection when processing quantum information.

Response

We acknowledge that this is an important point towards the full manipulation of the encoded qubits, which we have not discussed beyond adiabatic manipulation. We are currently working on this point by studying perturbations that select particular dark states in the degenerate manifold. We feel that the results presented here, concerning the connection between symmetry and dimensionality of the DDS, engineering states within the protected space, and optimizing purity and fidelity are important enough to warrant publications. We are confident that our result will stimulate further work, also concerning readout platforms of the processed information.

REVIEWERS' COMMENTS:

Reviewer #1 (Remarks to the Author):

In their reply the authors argue that my main criticisms on the experimental feasibility and an analysis of how to construct quantum information devices based on the theoretical framework presented in the paper are beyond the scope of this work and that is unreasonable to ask them to do this. However, these criticisms were not only based on my own expectations of what this paper should achieve but instead what it claims to have achieved. For instance, the sentence: "This approach offers new possibilities for storing, protecting and manipulating quantum information in open systems." in the abstract clearly indicates that the manuscript will discuss these topics and e.g. provide details about how to implement two qubit gates in their proposed setup. The title containing: "towards dissipation protected qubits" also indicates that this manuscript does provide a viable path towards quantum devices based on their ideas. For this to be the case one would at least expect a paper to describe how the DiVincenzo criteria can be fulfilled.

Numerous ideas and proposals for how to engineer qubits and process quantum information on various platforms and with different encodings are published on a regular basis. I consider for any of them to stand out and attract interest from a broad readership they need to be able to provide and discuss a feasible pathway towards a scalable quantum device with a realistic prospect of outperforming current and well established devices. I do not see this in the current manuscript and thus cannot recommend its publication in Nature Communications.

Reviewer #2 (Remarks to the Author):

I appreciate the authors' response to my notes and continue to recommend publication. My only additional comment is that I agree with the authors in their response to referee 1 that the referee is asking for too much from a theory paper. The current paper is sufficient in terms of new theoretical and experimental ideas to be published in a high-profile journal. The question of whether it is too specialized is more subjective - I think it is not.

Reviewer #3 (Remarks to the Author):

The authors have addressed my questions. I would like to recommend publication of this work in Nature Communications.